# Effects of In-Channel Structure on Chinook Salmon Spawning Habitat and Embryo Production

Robyn L. Bilski [1,*], Joseph M. Wheaton [2] and Joseph E. Merz [3,4]

1   California Department of Fish and Wildlife, West Sacramento, CA 95605, USA
2   Department of Watershed Sciences, Utah State University, Logan, UT 84322, USA; joe.wheaton@usu.edu
3   Cramer Fish Sciences, River Science and Restoration Lab., West Sacramento, CA 95691, USA; jmerz@fishsciences.net
4   Department of Ecology & Evolutionary Biology, University of California, Santa Cruz, CA 95060, USA
*   Correspondence: robyn.bilski@wildlife.ca.gov

**Abstract:** Adult salmonids are frequently observed building redds adjacent to in-channel structure, including boulders and large woody debris. These areas are thought to be preferentially selected for a variety of reasons, including energy and/or predation refugia for spawners, and increased hyporheic exchange for incubating embryos. This research sought to quantify in-channel structure effects on local hydraulics and hyporheic flow and provide a mechanistic link between these changes and the survival, development, and growth of Chinook salmon *Oncorhynchus tshawytscha* embryos. Data were collected in an eight-kilometer reach, on the regulated lower Mokelumne River, in the California Central Valley. Nine paired sites, consisting of an area containing in-channel structure paired with an adjacent area lacking in-channel structure, were evaluated. Results indicated that in-channel structure disrupts surface water velocity patterns, creating pressure differences that significantly increase vertical hydraulic gradients within the subsurface. Overall, in-channel structure did not significantly increase survival, development, and growth of Chinook salmon embryos. However, at several low gradient downstream sites containing in-channel structure, embryo survival, development, and growth were significantly higher relative to paired sites lacking such features. Preliminary data indicate that adding or maintaining in-channel structure, including woody material, in suboptimal spawning reaches improves the incubation environment for salmonid embryos in regulated reaches of a lowland stream. More research examining temporal variation and a full range of incubation depths is needed to further assess these findings.

**Keywords:** in-channel structure; large woody debris; vertical hydraulic gradient; Chinook salmon; spawning habitat; embryo production; California Central Valley

## 1. Introduction

Habitat heterogeneity is thought to be positively correlated with biotic production and species diversity [1–3]. In riverine systems, in-channel structures (e.g., large woody debris, bank irregularities, bedrock outcrops, roots, and boulders) play an important role in maintaining habitat diversity, increasing organic matter retention, and inducing changes in channel morphology [4,5]. In-channel structures change channel morphology by promoting sediment scour and aggregation in alluvial streams and controlling and maintaining the formation of channel features [6–9], although flow regime and sediment supply must be adequate to maximize benefits [10].

In-channel structure, such as woody material (e.g., root wads, branches, and tree trunks) and boulders, disrupts the hydraulic flow field, forcing otherwise more uniform flow patterns to diverge around structures, causing convergent flow patterns adjacent to and downstream of such structures, and often producing large eddies and secondary flow cells in their wake [11,12]. These hydraulic responses can be exacerbated at higher flows

leading to the forcing of pools in areas of convergent flow, formation of bank-attached and mid-channel bars in areas of divergent flow, and provision of secondary flow cells (eddies) as high flow energy refugia.

Cover, and hydraulic and geomorphic diversity associated with in-channel structures have been shown to provide a variety of functional benefits to salmon and trout (Salmonidae). In-channel structure provides important habitat for juvenile salmonids, offering protective cover from predators, high flows and solar energy and reduce aggressive interaction between juveniles that may influence energy reserves during crucial developmental stages [13–16]. Additionally, in-channel structure appears to have a significant effect on natural reproduction of salmonids. House [6] and Buffington et al. [17] found that channel roughness elements have significant impacts on river channel morphology and can trap sediments suitable for spawning. Furthermore, adult salmonids have effectively utilized gravels that accumulate adjacent to large woody debris or boulders to build redds and a strong association between Chinook salmon *Oncorhynchus tshawytscha* redds and large woody material has been established in marginal habitats [18,19]. In some river reaches, spawning has substantially increased in response to large woody debris placement [20]. It is hypothesized that in-channel structures may provide resting areas, cover, and visual barriers for breeding adult salmon, reducing stress during the spawning process [18,21].

Spawning salmonids may also be attracted to areas containing in-channel structure due to surrounding water velocity patterns, which extend into the subsurface, providing developing embryo benefits. Previous research has shown that concave bedforms cause changes in hyporheic flow [22–24] and hyporheic water temperatures are modified by hummocks on the stream-bed surface [25]. Crispell and Endreny [26] inferred from hydraulic simulations and temperature monitoring that in-channel structures modify hyporheic exchange flow. According to Esteve [27], female salmon perform exploratory behaviors during spawning site selection, suggesting they actively evaluate environmental conditions. Groundwater–surface water interactions and associated hyporheic water quality can influence salmonid embryos survival [28,29] and may have a direct impact on spawning site selection [30,31].

Although past research has attempted to elucidate the connection between hyporheic exchange and associated water quality with increased salmonid embryo survival, the relationship between structure, such as large woody debris and hyporheic flow patterns has received much less attention [25,32].

The hydraulic and geomorphic influences of in-channel structure are well documented, and those responses (e.g., creation of concave bedforms) have been shown in other circumstances to increase rates of hyporheic exchange [33,34]. It is, thus, logical that some empirical evidence linking the presence of in-channel structure to a hyporheic response might better substantiate or refute the conjecture that in-channel structures not only promote better salmonid spawning habitat, but also better embryo survival. Moreover, given the prevalence of placing large woody debris in flowing waters to increase habitat heterogeneity for the benefit of fish, there is a pressing need to better understand these links.

The purpose of this paper is to report direct empirical evidence related to whether in-channel structure alone can promote salmon embryo benefits. Specifically, we test the hypothesis that structural complexity, in the form of large woody debris and boulders, has a significant effect on hyporheic flow through the egg pocket, affecting hyporheic water quality and in turn, salmon embryo survival, development, and growth. This study is most concerned with whether there is empirical evidence to support this hypothesis, which can then subsequently establish whether there is a need for studies that focus more explicitly on the mechanistic links between these different processes.

## 2. Materials and Methods

### 2.1. Study Area

The Mokelumne River is a modified, snow fed system that drains approximately 1700 km$^2$ of the Central Sierra Nevada Mountain Range. The lower Mokelumne River

(LMR) extends from the base of Camanche Dam (river kilometer (rkm) 103), a non-passable structure to anadromous fishes, to its confluence with the San Joaquin River (rkm 0) in the California Bay-Delta. Prior to the development of Pardee Reservoir (rkm 119; 0.24 km$^3$) in 1928, annual peak LMR flows averaged 263 m$^3$/s [35]. After the development of Camanche Reservoir (0.51 km$^3$) in 1963, annual average peak flows were reduced to 54 m$^3$/s [36]. Due to these developments, the LMR has undergone many changes. A thin band of riparian vegetation has moved into the formerly active channel, floodplain vegetation has been replaced with agricultural fields, the active channel has been reduced to one-half its previous width, and bed sediments are less mobile, reducing the quantity and quality of anadromous salmonid spawning and incubation habitat [35,36]. Currently, the LMR supports two species of native anadromous salmonids, fall-run Chinook salmon and winter-run steelhead *O. mykiss* [37]. The majority of Chinook salmon spawning occurs within an 8 km reach of the LMR, beginning just below the base of Camanche Dam [18]. Project data were collected within nine paired sites along this spawning reach (Figure 1). Three paired sites were chosen within upstream, midstream, and downstream segments of the spawning reach to account for data replication [38].

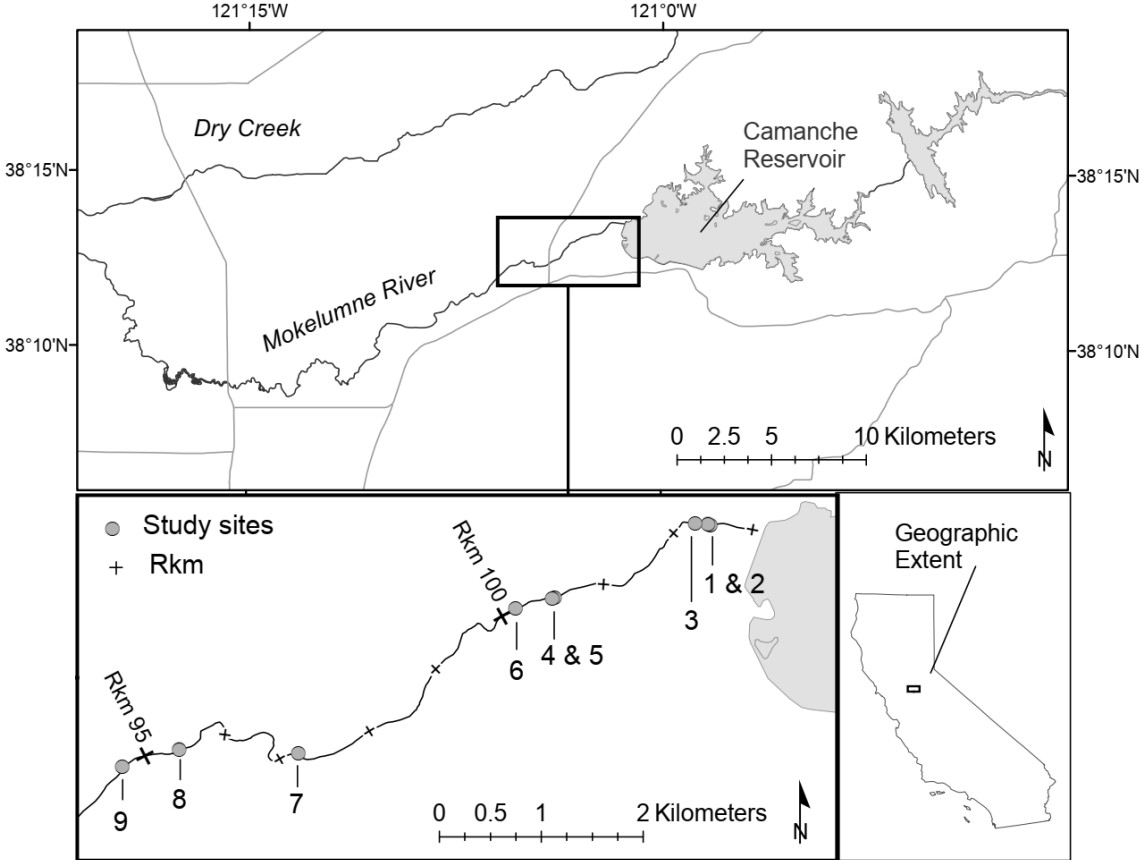

**Figure 1.** Study sites (1–9) on the lower Mokelumne River, California. Each site included a pair of independent spawning areas with similar geomorphic units; one containing in-channel structure and one without.

Sediment size, river gradient, and land use activities along the riparian corridor varied among the nine-paired sites. Upstream Sites 1–3 were located near the confluence with Murphy Creek, a small rain-fed tributary [39] and flanked by a large recreational day-use area and native and exotic riparian vegetation. Midstream and downstream Sites 4–9 were adjoined by livestock pastures and/or nearby agricultural crop fields and bordered by native and exotic riparian vegetation. Recent bed surface and subsurface sediment samples, collected within the upper 13 km of the LMR, indicated bed material fining occurs in a

downstream progression from Site 1 to Site 9 [40,41]. Core samples taken during these studies indicated that the median particle diameter at 50% in the cumulative distribution ($D_{50}$) ranged between 35.2 mm and 45.6 mm and averaged 38.9 (SD = 4.0) in the upstream reaches of the LMR (rkm 100–103). The $D_{50}$ ranged between 18.4 and 38.9 and averaged 25.6 (SD = 6.6) in the downstream LMR reaches (rkm 95–100). Similar to grain size, river gradient was also inversely correlated with distance from Camanche Dam and decreased from Site 1 (0.91 m/km) through Site 9 (0.11 m/km) [18,40,41]. Spawning gravels, large woody debris, and boulders had been added to many of the upstream and midstream sites as part of annual spawning habitat improvement projects on the LMR over the last 15 years (Table 1). In contrast, most of the downstream sites had not been enhanced with spawning gravels and were located within a river segment containing a significantly higher proportion of fine materials and reduced gravel permeability, relative to upstream segments of the spawning reach [40,42].

**Table 1.** Descriptions of paired study sites on the lower Mokelumne River, California (LWD = large woody debris). The number of structures at each site are provided in the second column.

| Site Number | Form of In-Channel Structure | River Kilometer | Year(s) Enhanced |
|---|---|---|---|
| 1 | 1 Boulder | 102.5 | 1999, 2005 |
| 1 | Control | 102.5 | 1999, 2005 |
| 2 | 1 LWD | 102.4 | 1999, 2005 |
| 2 | Control | 102.4 | 1999, 2005 |
| 3 | 1 Boulder | 102.3 | 1992, 1993, 2006 |
| 3 | Control | 102.2 | 1992, 1993, 2006 |
| 4 | 2 Boulders | 100.5 | 2002 |
| 4 | Control | 100.5 | 2002 |
| 5 | 2 Boulders | 100.4 | 2002 |
| 5 | Control | 100.4 | 2002 |
| 6 | 1 LWD | 100.1 | None |
| 6 | Control | 100.3 | None |
| 7 | 1 LWD | 97.2 | None |
| 7 | Control | 97.2 | None |
| 8 | 1 LWD | 95.4 | None |
| 8 | Control | 95.4 | None |
| 9 | 1 Boulder | 94.8 | 1997 |
| 9 | Control | 94.8 | 1997 |

*2.2. Site Selection*

The experiment focus was to compare survival, development, and growth (as indicated by length) of Chinook salmon embryos within and away from the direct influence of in-channel structure. To do this we focused on paired sites, which were selected to represent spawning areas encompassing and deficient of in-channel structure, in the form of large woody debris and boulders. Each study site was approximately 3 to 5 m in length and each site pairing consisted of one site containing in-channel structure and another nearby control site (within 5 m), having similar characteristics (geomorphic unit and position in the river channel), lacking in-channel structure. To be defined as a site with in-channel structure, they needed to contain at least one in-channel structure feature (Table 1). Qualifying boulders measured 60 to 120 cm in diameter and weighed between 250 kg and 500 kg [8]. Qualifying large woody debris had a minimum diameter of 10 cm and measured over 2 m in length [43]. Due to the variable localized velocity patterns detected around flow deflectors (e.g., large woody debris and boulders) [44], distinct measurement locations were designated just upstream, downstream, and lateral of in-channel structure and control areas at each site (Figure 2). Measurement locations at sites containing in-channel structure were placed an average of 0.5 m (SD = 0.2) from the objects.

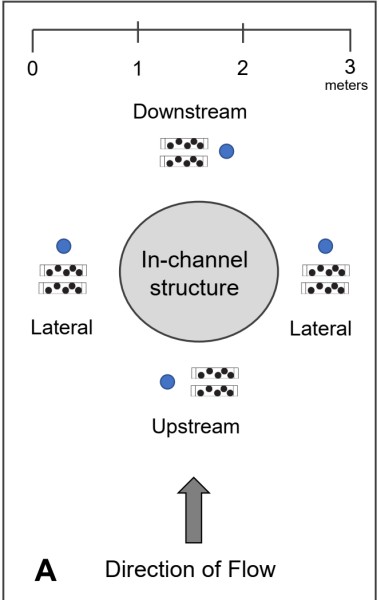
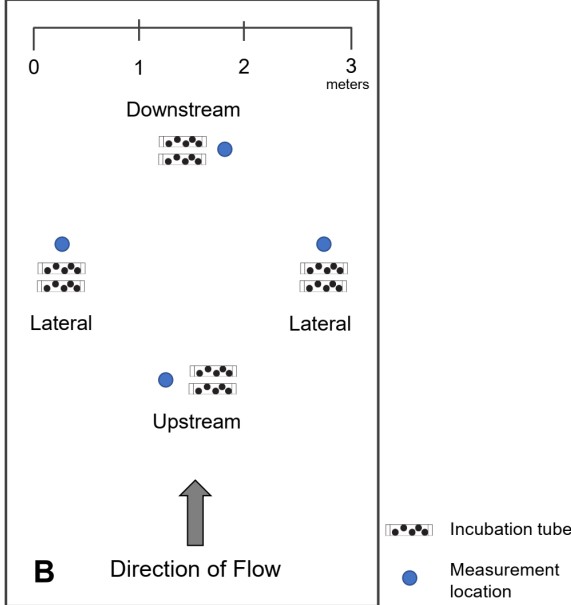

**Figure 2.** Schematic of measurement locations (upstream, lateral, and downstream) and incubation tubes designated around paired sites (**A**) containing in-channel structure and (**B**) lacking in-channel structure, on the lower Mokelumne River, California. A total of 8 egg tubes was placed around each site, two at each measurement location. Incubation tubes not drawn to scale.

### 2.3. Hydraulics—Surface Water Depth and Velocity and River Flow

To determine whether in-channel structure influenced hydraulics in a consistent fashion, as suggested by past investigators [33], we measured depths and velocity profiles at each measurement location. Surface water depths and velocities were measured using a top setting wading rod and a Marsh McBirney model 511 velocity meter, which uses an electromagnetic sensor to read subtle positive and negative velocity measurements on a vertical ($y$) and horizontal ($x$) axis. Water velocities were recorded every 0.1 m along the water column at each measurement location (2 to 11 per location, varying with depth) and then depth averaged. These data were collected once at each measurement location, when hourly flow released at Camanche Dam (LMR rkm 103) was stable and ranged from 9.6 m³/s to 9.7 m³/s. These measurements took place shortly after embryo incubation, within a three-day period. It was necessary to collect these data during a period when river flow was stable, so comparisons could be made between and among paired sites.

During the entire study period (late fall through winter), average daily flow (LMR rkm 103) ranged from 9.6 m³/s to 9.9 m³/s and hourly flow ranged from 9.5 m³/s to 12.8 m³/s [45]. The largest change in daily average outflow below Camanche Reservoir was 0.3 m³/s and the largest change in hourly outflow was 3.2 m³/s. Changes in hourly outflow that exceeded 0.1 m³/s took place on just 6 instances during the experiment.

### 2.4. Hyporheic Flow (Vertical Hydraulic Gradient)

We were particularly interested in testing whether hyporheic flows were influenced by in-channel structure. To measure hyporheic flows we used mini piezometers, which were constructed using a 3 mm diameter polyethylene tube and a sampling tip approximately 1.5 cm wide by 3 cm high [46]. The stainless-steel tip draws hyporheic water from a 1 cm screened section and connects to the surface through the attached polyethylene tube. The piezometer tubes emerged approximately 10–30 cm from the gravel surface and were plugged with golf tees to prevent surface water intrusion between sampling periods. Piezometer installation was performed using a steel drive rod to hammer each monitoring point into the subsurface. Each piezometer tip was installed approximately 22 cm below the gravel, falling within the range of measured egg burial depths for Chinook salmon in California [47]. The same burial depth was used in a previous egg incubation

study on the LMR [40]. Hyporheic flow data were collected at each measurement location in the surrounding hyporheic water, just adjacent (0.3 m) to the embryo incubation tubes (Figure 2). The vertical hydraulic gradient was measured by pumping surface and hyporheic water into a bubble manometer board, an instrument that compares water pressure differences between the river and shallow depths in the gravel bar [46]. Upwelling conditions were suggested by a positive measurement, whereas negative measurements indicated downwelling. Vertical hydraulic gradient measurements were recorded two weeks after piezometer installation to allow adequate time for the gravels to resettle around the mini piezometers [48]. Measurements were collected at the beginning (early December) and towards the end (late December) of when we expected all embryos would have hatched (see below). Hyporheic flow data were measured twice and averaged at each measurement location during a sample period.

### 2.5. Hyporheic and Surface Water Quality

Dissolved oxygen, pH, water temperature, and conductivity measurements were taken given the importance of these parameters during early Chinook salmon embryo development [49–52]. Lower pH levels within the subsurface may identify areas with less hyporheic exchange due to the breakdown of organic material [46]. Hyporheic water conductivity measurements were used to help indicate the presence of long residence groundwater which contains more dissolved ions than water that is rapidly flushed through the subsurface due to increased mineral and organic matter contact [46,53].

Intergravel water quality measurements were taken by connecting mini piezometers to a peristaltic pump. Subsurface water was pulled into an enclosed flow-through chamber to prevent atmospheric oxygen from altering hyporheic dissolved oxygen levels and to provide adequate water flow past the dissolved oxygen sensor [51]. Surface and hyporheic water conductivity, pH, and dissolved oxygen readings were recorded using Orion 128, 210A, and YSI 550 m, respectively. Surface water temperature readings were recorded using a YSI 550 m. Meter tips were inserted into an airtight flow-through chamber using rubber adapters. Prior to sampling, water quality meters were calibrated in a laboratory. In addition, the meters were calibrated in the field daily. Water in the flow-through chamber was emptied after each sample was taken.

Conductivity, pH, and dissolved oxygen measurements were taken in conjunction with vertical hydraulic gradient measurements at the beginning (early December) and towards the end (late December) of the Chinook salmon embryo incubation period. Surface water temperature readings were also collected at these intervals. This was done to account for some of the variation found in hyporheic flow and water quality during the spawning and incubation season. Hyporheic flow and water quality data were measured twice and averaged at each measurement location during a sample period. A total of eight measurements ($n = 8$) were recorded at each site and a total of sixteen measurements ($n = 16$) were recorded at each paired site. Average daily flow below Camanche Reservoir ranged between 9.6 m$^3$/s and 9.8 m$^3$/s and averaged 9.7 m$^3$/s during the time frame surrounding hyporheic flow and water quality measurements and embryo incubation.

Intergravel water temperatures were recorded hourly, each day, during the 29-day embryo incubation period using StowAway Tidbit waterproof temperature loggers (Onset Computing). The loggers were attached to one embryo incubation tube per measurement location and buried 22 cm below the gravel surface. The average daily temperature during the 29-day incubation period was calculated at each measurement location and used for analysis.

### 2.6. Embryo Survival, Development, and Growth

To test the hypothesis that structural complexity influences hyporheic flow through the egg pocket, and hence, salmon embryo survival, development, and growth, we exposed Chinook salmon embryos to in-channel structure and control locations at each site (Figure 2). Embryos used for the incubation treatment were acquired from adult fall-run Chinook

salmon that had returned to the Mokelumne River Fish Installation (LMR rkm 103). The fertilized eggs were produced in late November by spawning approximately 40 female Chinook salmon with approximately 40 males at a one-to-one ratio and mixed to account for parental differences. All eggs were disinfected for 20 min in an iodine solution [54]. A protocol for the humane care and use of live animals was completed prior to the study. The number of fertilized eggs used for the study was reduced to the level required for statistical validation based on the results of a previous experiment [35].

The embryo incubation tubes measured 305 mm in length, 44.5 mm in diameter and were constructed from polyvinyl chloride pipes (grade 35) with a polyvinyl chloride cap on each end [40]. We followed the methods of a successful study by Merz et al. [40] analyzing the growth and survival of salmonid embryos. Approximately 200 eggs were placed inside each incubation tube at the hatchery in early December and three tubes remained at the hatchery (Hatchery Control Group). The other tubes were transported in buckets filled with river water to the study sites. One artificial redd consisting of two incubation tubes was constructed at each measurement location, just adjacent (0.3 m) to the mini piezometer. Each paired Site consisted of 16 incubation tubes; 8 tubes were placed around the control area and 8 tubes were placed around the in-channel structure (Figure 2). We created artificial redds by digging depressions approximately 22 cm deep using a metal rake. Similar to natural redd construction, we cleaned fine sediment adjacent to the egg pocket and developed a tail berm just downstream to mimic spawning female activity and reflect natural redd conditions. Two tubes were placed inside the depression side by side and positioned horizontally, perpendicular to the river flow. Each redd was then backfilled with gravel from just upstream of the depression to replicate natural redd construction [47].

We selected a 29-day incubation period for the 14-day-old Chinook salmon embryos to compare their early life stage development (hatching rate and alevin growth) anticipating hyporheic water temperatures would range from 10 to 13 °C [40,55]. The tubes were recovered from the river and hatchery in early January. A crew of four to five people removed tubes from each measurement location to determine the survival and hatching rate of all embryos on the same day. Immediately after removal from the gravel, tubes were placed in a bucket of river water and transported to the shore for processing. Living and dead embryos were emptied from each tube and sorted in a shallow tray. Embryo survival and the level of development (egg vs. alevin) were recorded. Up to 50 living alevins from each tube were placed in a labeled bag filled with ethanol on site and transported to a laboratory where total length of each alevin was measured to the nearest millimeter. Beyond the 50 living embryos taken from each egg tube, the remaining surviving embryos were released into artificial redds in the river after the study period.

*2.7. Statistical Analyses*

Data distributions were tested for normality using the Shapiro–Wilk test to determine if parametric or nonparametric statistical tests were appropriate. A three-way analysis of variance (ANOVA) was used to determine if there were significant differences in surface water velocity and vertical hydraulic gradient at sites containing in-channel structure, relative to sites that lacked in-channel structure. The presence/absence of in-channel structure, measurement location (upstream, downstream, and lateral), and Site number were the main factors for each three-way ANOVA. Because substrate size, channel gradient, and water temperature were found to correlate with distance from Camanche Dam (18, 40, 41), we used Site number as a dummy variable to encompass these relationships. The interaction between the presence/absence of in-channel structure and measurement location and the interaction between the presence/absence of in-channel structure and Site number were also examined.

Because upwelling and downwelling measurements may cancel each other out, and because vertical hydraulic gradient magnitude and direction are both important [53,56], two separate analyses were run for vertical hydraulic gradient. Directional vertical hydraulic gradient values, positive (upwelling) and negative (downwelling), were used for

one analysis. The absolute value of each vertical hydraulic gradient measurement was calculated to analyze vertical hydraulic gradient magnitude. A Box–Cox transformation was used to transform vertical hydraulic gradient magnitude data to a normal distribution. Transformations to a normal distribution were unsuccessful for embryo survival, development, and growth data. To determine if there were within-site differences in embryo survival, development, and growth between areas containing and lacking in-channel structure, we used a Wilcoxon rank-sum test.

Generalized Linear Models (GLM) were used to analyze the relationships between physical and chemical habitat parameters and embryo survival, development, and growth. For rate data (survival rate and hatching rate) GLMs with the Poisson distribution were used. The following independent variables were analyzed: (1) surface water depth (m), (2) total surface water velocity (m/s), (3) horizontal surface water velocity (m/s), (4) vertical surface water velocity (m/s), (5) hyporheic dissolved oxygen (mg/L), (6) hyporheic pH, (7) hyporheic conductivity (μS/cm), (8) average daily hyporheic water temperature (°C), (9) vertical hydraulic gradient magnitude, and (10) vertical hydraulic gradient (directional). We built a correlation matrix to examine the relationships between independent variables. Variables having a high level of collinearity with each other were not used in the same models. The final models and associated independent variables were selected based on the lowest Akaike information criterion (AIC) score.

Statistical tests were completed using JMP 16.0.0 (SAS Institute, Inc., Cary, NC, USA). A *p*-Value of $\leq 0.05$ was considered a statistically significant result.

## 3. Results

### 3.1. Hydraulics—Surface Water Velocity

Measurement location, site number, and structure presence/absence had a statistically significant effect on horizontal water velocity (Table 2). The interaction between the main effects (measurement location and in-channel structure presence/absence) also had a statistically significant effect on horizontal water velocity (Table 2). Specifically, horizontal water velocities at downstream measurement locations of sites containing in-channel structure were significantly lower than water velocities at all other measurement locations (Figure 3A). In addition, the presence of in-channel structure increased horizontal water velocities at lateral measurement locations, in comparison to the upstream and downstream measurement locations at sites containing in-channel structure (Figure 3A). In contrast, sites lacking in-channel structure showed little variation between water velocities at the upstream, downstream, and lateral measurement locations (Figure 3A).

**Table 2.** Three-way analysis of variance results of vertical and horizontal surface water velocity on the lower Mokelumne River. The effects of site number (1–9), in-channel structure presence (Yes versus No), measurement location (upstream, lateral, and downstream) and interactions of the main effects were tested. Bold indicates statistical significance (*p* < 0.05).

| Source | Sum of Squares | df | *F* | *p*-Value |
|---|---|---|---|---|
| Vertical surface water velocity | | | | |
| Main effects | | | | |
| Site number | 0.045 | 8 | 1.93 | 0.089 |
| Structure presence | 0.019 | 1 | 6.70 | **0.014** |
| Measurement location | 0.061 | 2 | 10.56 | **0.000** |
| Interactions | | | | |
| Structure presence · measurement location | 0.041 | 2 | 7.05 | **0.003** |
| Structure presence · site number | 0.018 | 8 | 0.76 | 0.641 |
| Error | 0.093 | 32 | | |
| Horizontal surface water velocity | | | | |
| Main effects | | | | |
| Site number | 2.836 | 8 | 5.59 | **0.000** |

**Table 2.** *Cont.*

| Source | Sum of Squares | df | F | p-Value |
|---|---|---|---|---|
| Structure presence | 0.317 | 1 | 5.00 | **0.032** |
| Measurement location | 2.260 | 2 | 17.83 | **<0.001** |
| Interactions | | | | |
| Structure presence · measurement location | 1.618 | 2 | 12.76 | **<0.001** |
| Structure presence · site number | 0.580 | 8 | 1.14 | 0.362 |
| Error | 2.028 | 32 | | |

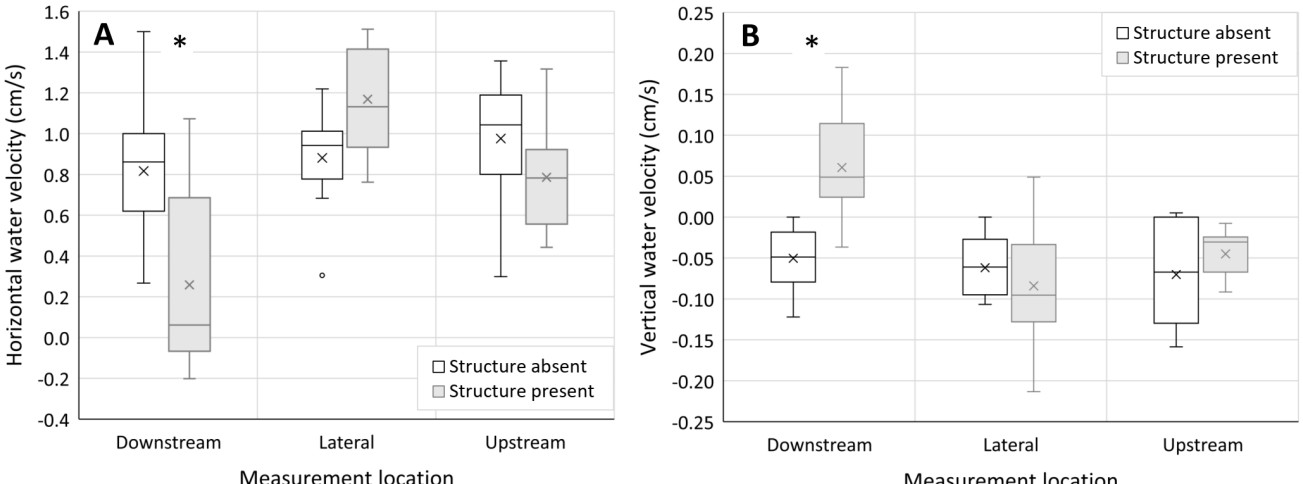

**Figure 3.** Box and whisker plots of (**A**) horizontal and (**B**) vertical surface water velocities at measurement locations (downstream, lateral, and upstream) where in-channel structure was present (grey plots) versus absent (open plots). The box represents the data range between 1st and 3rd quartile, the horizontal line in the box represents the median, the x in the box represents the mean, and the whiskers represent the minimum and maximum values excluding the outliers (small circles). Asterisks (*) indicate a significant difference between measurement locations containing in-channel structure and lacking in-channel structure.

### 3.2. Hyporheic Flow

Measurement location and the interaction between in-channel structure presence/absence and site number had a statistically significant effect on directional vertical hydraulic gradient (Table 3). The interaction between in-channel structure presence/absence and measurement location also had a statistically significant effect on directional vertical hydraulic gradient (Table 3). The upstream measurement locations at sites containing in-channel structure showed evidence of downwelling, or the movement of water from the surface into the subsurface and were significantly different than the directional vertical hydraulic gradient at all other measurement locations (Figure 4). Conversely, upwelling of hyporheic water was evident at downstream measurement locations at sites containing in-channel structure, as the mean directional vertical hydraulic gradient was positive and greater than the mean directional vertical hydraulic gradient at all other measurement locations (Figure 4). The presence of in-channel structure significantly increased vertical hydraulic gradient magnitude relative to sites lacking in-channel structure (Table 3).

### 3.3. Embryo Survival, Development, and Growth

Chinook salmon embryo survival rate was inconsistent between sites, including the hatchery control site. Hatchery Control Group survival rates ranged from 4–48% while in-river survival ranged from 0–58% (Sites 1–9). Mean Hatchery Control Group survival was 27% (Standard error (SE) = 13). Mean survival within upstream Sites 1–3 was 6% (SE = 2), which was relatively low when compared with mean survival rates of 25% (SE = 3)

at midstream Sites 4–6, and 23% (SE = 4) at downstream Sites 7–9. Statistical tests for within-site differences indicated that Chinook salmon embryo survival rates were significantly higher at measurement locations containing in-channel structure, relative to those lacking in-channel structure, within downstream Sites 7 and 9 (Table 4; Figure 5A). Sample variance for embryo survival between the incubation tubes used for the experiment was 4.2% (*n* = 129).

**Table 3.** Three-way analysis of variance results of vertical hydraulic gradient on the lower Mokelumne River. The effects of site number (1–9), in-channel structure presence (Yes versus No), measurement location (upstream, lateral, and downstream) and interactions of the main effects were tested. Bold indicates statistical significance (*p* < 0.05).

| Source | Sum of Squares | df | *F* | *p*-Value |
|---|---|---|---|---|
| Directional vertical hydraulic gradient | | | | |
| Main effects | | | | |
| Site number | 2.936 | 8 | 1.72 | 0.131 |
| Structure presence | 0.201 | 1 | 0.94 | 0.339 |
| Measurement location | 6.909 | 2 | 16.20 | **<0.0001** |
| Interactions | | | | |
| Structure presence · measurement location | 10.648 | 2 | 24.97 | **<0.0001** |
| Structure presence · site number | 8.601 | 8 | 5.04 | **0.000** |
| Error | 6.823 | 32 | | |
| Vertical hydraulic gradient magnitude | | | | |
| Main effects | | | | |
| Site number | 0.883 | 8 | 0.89 | 0.540 |
| Structure presence | 1.662 | 1 | 13.33 | **0.001** |
| Measurement location | 0.500 | 2 | 2.00 | 0.151 |
| Interactions | | | | |
| Structure presence · measurement location | 0.004 | 2 | 0.01 | 0.986 |
| Structure presence · site number | | 8 | 2.74 | **0.020** |
| Error | 3.991 | 32 | | |

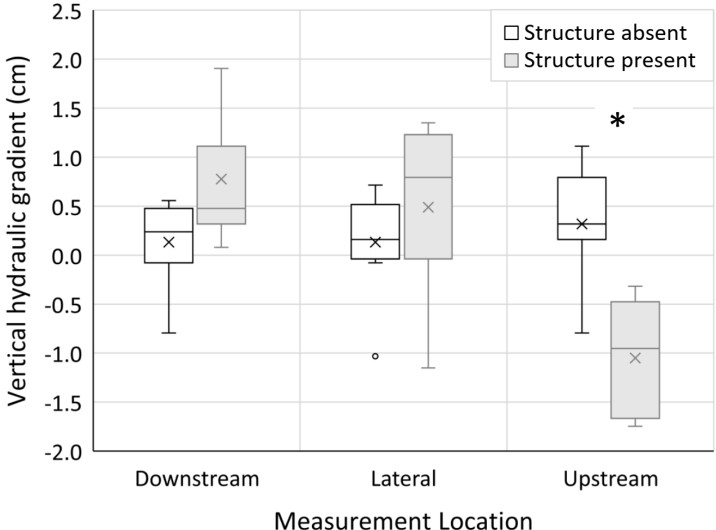

**Figure 4.** Box and whisker plots of directional vertical hydraulic gradient at measurement locations (downstream, lateral, and upstream) where in-channel structure was present (grey plots) versus absent (open plots). The box represents the data range between 1st and 3rd quartile, the horizontal line in the box represents the median, the x in the box represents the mean, and the whiskers represent the minimum and maximum values excluding the outliers (small circles). Asterisks (*) indicate a significant difference between measurement locations containing in-channel structure and lacking in-channel structure.

**Table 4.** Results of Wilcoxon rank-sum tests for within-site differences in embryo survival rate, hatching rate, and total length between areas containing and lacking (control) in-channel structure. Z represents the test statistic and S represents the total rank sum. Bold indicates statistical significance ($p < 0.05$).

| Site | S | Z | *p*-Value |
|---|---|---|---|
| *Embryo survival* | | | |
| 1 | 34.5 | −1.344 | 0.179 |
| 2 | 72.0 | 0.447 | 0.655 |
| 3 | 68.0 | 0.000 | 1.000 |
| 4 | 45.5 | −1.168 | 0.243 |
| 5 | 63.5 | −0.420 | 0.674 |
| 6 | 39.0 | 0.000 | 1.000 |
| 7 | 93.0 | 2.573 | **0.010** |
| 8 | 35.0 | −0.929 | 0.353 |
| 9 | 44.0 | 2.523 | **0.012** |
| *Embryo hatching rate* | | | |
| 1 | 36.0 | −1.097 | 0.273 |
| 2 | 78.0 | 0.998 | 0.318 |
| 3 | 77.0 | 0.893 | 0.372 |
| 4 | 45.0 | −1.216 | 0.224 |
| 5 | 75.0 | 0.683 | 0.495 |
| 6 | 43.0 | 0.560 | 0.575 |
| 7 | 91.0 | 2.363 | **0.018** |
| 8 | 34.0 | −1.071 | 0.284 |
| 9 | 45.0 | 2.710 | **0.007** |
| *Embryo total length* | | | |
| 1 | 212.0 | −2.307 | **0.021** |
| 2 | 1219.0 | 0.521 | 0.603 |
| 3 | 7337.5 | −1.268 | 0.205 |
| 4 | 32,127.5 | 2.490 | **0.013** |
| 5 | 101,812.0 | 5.948 | **<0.0001** |
| 6 | 6019.5 | −4.427 | **<0.0001** |
| 7 | 21,444.0 | −1.919 | 0.055 |
| 8 | 97,459.0 | −6.370 | **<0.0001** |
| 9 | 102.0 | −2.356 | **0.019** |

Chinook salmon egg hatching rates in the Hatchery Control Group ranged from 64 to 75% and were less variable than those at in-river Sites 1–9 (range = 0–66%). The mean in-river hatching rate was 44% (SE = 3) at upstream Sites 1–3, 35% (SE = 4) at midstream Sites 4–6, and 28% (SE = 4) at downstream Sites 7–9. Similar to survival trends, Chinook salmon egg hatching rates were significantly higher at measurement locations containing in-channel structure, relative to those lacking in-channel structure, within downstream Sites 7 and 9 (Table 4; Figure 5B). Sample variance for embryo hatching rates between the incubation tubes used for the experiment was 5.8% (*n* = 129).

The total lengths of Hatchery Control Group Chinook salmon alevins ranged from 23–26 mm. The size of alevins incubated in the river was more variable, as total lengths ranged from 12–26 mm. Chinook salmon alevins recovered at upstream Sites 1–3 had the highest mean total length of 22.3 mm (SE = 0.1). Mean total length decreased to 21.8 mm (SE = 0.05) within midstream Sites 4–6 and 21.1 mm (SE = 0.05) within downstream Sites 7–9. Within Site 1, alevin total length at measurement locations lacking in-channel structure was significantly higher than alevin total length at measurement locations containing in-channel structure (Table 4; Figure 5C). However, within Sites 4, 5, 6, 8, and 9, the total length of alevins at measurement locations containing in-channel structure was significantly higher than the total length of alevins at measurement locations lacking in-channel structure (Table 4; Figure 5C).

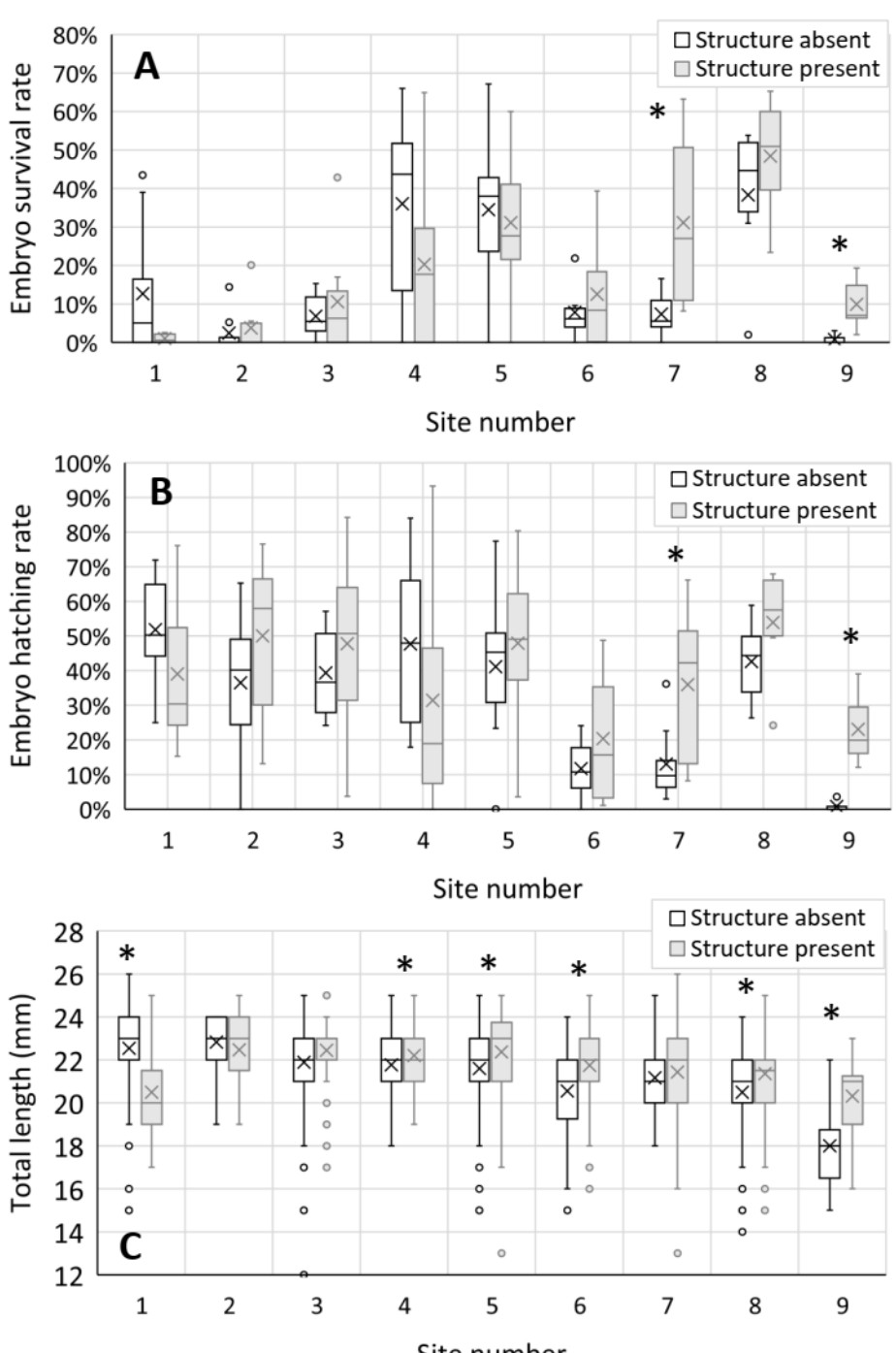

**Figure 5.** Box and whisker plots of Chinook salmon embryo (**A**) survival, (**B**) hatching rate, and (**C**) total length by site number at sites containing (grey plots) versus lacking (open plots) in-channel structure (control). The box represents the data range between 1st and 3rd quartile, the horizontal line in the box represents the median, the x in the box represents the mean, and the whiskers represent the minimum and maximum values excluding the outliers (small circles). Asterisks (*) indicate a significant within site difference.

*3.4. Physical and Chemical Conditions Associated with Embryo Survival, Development, and Growth*

Mean hyporheic pH, conductivity, and water temperature measurements were similar to stream (surface) water values at most of the sites (Table 5). However, there was a considerable increase in mean hyporheic conductivity at Site 6, where in-channel structure was absent. Average hyporheic dissolved oxygen values at sites containing and lacking in-

channel structure were generally lower than stream water dissolved oxygen levels (Table 5). Hyporheic dissolved oxygen values at measurement locations varied considerably, ranging from 0.9 to 12.0 mg/L. The range of hyporheic pH values recorded at measurement locations fell between 5.7 and 7.2. The minimum average daily intergravel water temperature of 10.5 °C was recorded on day 29 of the incubation period, while the maximum average daily intergravel water temperature of 13.4 °C was recorded on day 3 of incubation.

**Table 5.** Mean physical and chemical parameters associated with sites 1–9 on the lower Mokelumne River. Abbreviations are as follows: DO = dissolved oxygen; COND = conductivity; TEMP = average daily temperature; VHG (M) = vertical hydraulic gradient (magnitude). Standard deviations are in parentheses.

| Sites →/ Parameters ↓ | 1 | 2 | 3 | 4 | 5 | 6 | 7 | 8 | 9 |
|---|---|---|---|---|---|---|---|---|---|
| Surface water | | | | | | | | | |
| DO (mg/L) | 9.4 | 9.6 | 8 | 10.2 | 9 | 10.2 | 8.9 | 9.2 | 10.4 |
| pH | 6.6 | 6.4 | 6.7 | 6.8 | 6.7 | 7 | 6.8 | 6.8 | 6.8 |
| COND (µS/cm) | 38 | 37.6 | 37.4 | 38.3 | 38.7 | 37.9 | 38 | 37.7 | 37.9 |
| TEMP (°C) | 12.2 | 11.9 | 11.9 | 11.4 | 11.4 | 11.3 | 11.8 | 11.8 | 11.7 |
| Hyporheic water—Structure present | | | | | | | | | |
| DO (mg/L) | 8.2 (1.0) | 8.6 (0.6) | 8.1 (1.9) | 8.9 (1.6) | 9.7 (1.3) | 8.2 (2.6) | 6.7 (2.1) | 7.3 (1.0) | 8.5 (1.6) |
| pH | 6.7 (0.2) | 6.8 (0.1) | 6.7 (0.2) | 6.5 (0.2) | 6.8 (0.3) | 6.6 (0.2) | 6.5 (0.2) | 6.5 (0.1) | 6.5 (0.3) |
| COND (µS/cm) | 37.4 (0.5) | 37.5 (0.6) | 37.4 (0.6) | 38.2 (0.6) | 39.2 (2.0) | 36.5 (1.0) | 37.4 (1.9) | 37.8 (0.8) | 37.8 (0.7) |
| TEMP (°C) | 11.8 (1.0) | 12 (0.9) | 11.9 (0.9) | 11.9 (1.0) | 12 (1.0) | 11.9 (1.0) | 11.7 (1.0) | 11.2 (1.2) | 11.6 (1.0) |
| VHG (M) | 0.9 (0.7) | 1.2 (0.5) | 1 (1.1) | 1.2 (0.9) | 0.3 (0.3) | 0.8 (0.8) | 1.6 (0.7) | 0.9 (0.5) | 0.8 (0.4) |
| Hyporheic water—Structure absent | | | | | | | | | |
| DO (mg/L) | 8.5 (0.6) | 9.1 (0.7) | 8.7 (1.4) | 9.9 (0.7) | 6.1 (1.9) | 3 (2.2) | 7.8 (2.4) | 8.6 (0.6) | 6.7 (0.9) |
| pH | 6.8 (0.1) | 6.8 (0.2) | 6.8 (0.2) | 6.7 (0.2) | 6.4 (0.2) | 6.5 (0.1) | 6.2 (0.4) | 6.7 (0.1) | 6.5 (0.1) |
| COND (µS/cm) | 37.5 (0.2) | 37.7 (0.2) | 37.3 (0.2) | 38.1 (0.3) | 39.0 (1.9) | 58.5 (12.8) | 38.5 (2.7) | 37.2 (0.9) | 38.1 (1.2) |
| TEMP (°C) | 11.9 (0.9) | 12.0 (0.9) | 12.0 (0.9) | 11.9 (1.0) | 11.9 (1.0) | 11.8 (0.9) | 11.7 (1.0) | 11.6 (1.0) | 11.7 (1.1) |
| VHG (M) | 0.5 (0.5) | 0.6 (0.4) | 0.2 (0.2) | 0.5 (0.4) | 0.5 (0.2) | 0.7 (0.2) | 0.5 (0.2) | 0.7 (0.5) | 0.9 (0.2) |

Variables having correlation coefficients greater than 0.6 or less than −0.6 precluded use of those related variables in the same models (Table S1). The best fit GLM for Chinook salmon embryo survival rate found that hyporheic water temperature had a weak negative relationship with survival (Table 6). The GLM for Chinook salmon embryo hatching rate showed a weak negative relationship between hyporheic water temperature and hatching rate and a weak positive effect from pH (Table 6). In contrast, the GLM for Chinook salmon growth (total length) had more variables included in the model and several had statistically significant relationships with alevin total length (Table 6). EC had a significant negative relationship with alevin total length. pH had a significant positive relationship with alevin total length, while hyporheic water temperature and vertical surface water velocity had weaker positive relationships with alevin total length.

**Table 6.** Summary statistics from the final Generalized Linear Models (GLM) for Chinook salmon embryo survival, hatching rate, and growth on the lower Mokelumne River, based on physical and chemical habitat variables. Terms have been ordered by effect direction and strength showing estimated model coefficients, standard error (SE), chi square, and *p*-Values. The final model was selected based on the lowest AIC score. Term abbreviations are as follows: ATEMP = average daily temperature, SWV-V = surface water velocity (vertical), COND = conductivity. Pr > ChiSq = chi square test statistic. Bold indicates statistical significance ($p < 0.05$).

| GLM | AIC | Term | Estimate | SE | Chi Square | Pr > ChiSq |
|---|---|---|---|---|---|---|
| Embryo survival | 44.650 | | | | | |
| | | Intercept | 8.523 | 10.294 | 0.546 | 0.460 |
| | | ATEMP | −0.863 | 0.877 | 0.747 | 0.387 |
| Embryo hatching rate | 65.842 | | | | | |
| | | Intercept | −5.461 | 11.652 | 0.234 | 0.629 |
| | | pH | 1.017 | 1.233 | 0.722 | 0.396 |
| | | ATEMP | −0.189 | 0.886 | 0.043 | 0.836 |
| Embryo growth | 131.528 | | | | | |
| | | Intercept | −1.772 | 8.056 | 0.048 | 0.826 |
| | | pH | 2.759 | 0.774 | 11.067 | **0.001** |
| | | SWV-V | 3.751 | 2.294 | 2.590 | 0.108 |
| | | ATEMP | 0.680 | 0.622 | 1.176 | 0.278 |
| | | COND | −0.071 | 0.034 | 4.210 | **0.040** |

## 4. Discussion

The presence of in-channel structure significantly increased variation in the surrounding physical habitat. Surface water velocity measurements at sites not associated with in-channel structure were largely homogeneous. In contrast, water velocity measurements taken at sites containing in-channel structure were characterized by accelerations along the sides and just downstream of the structures, as well as zones of reduced velocity on the lee side of structures. These flow shear zones are the result of flow separation around in-channel structure. Some shear zones can be characterized as eddies (flow along and downstream of seam moves in opposite direction), whereas others are simply wakes (slower flow on the seam wake side). These findings support studies describing the forcing of shear zones by several forms of in-channel structure [5,44]. The presence of in-channel structure increases hydraulic condition variability, potentially alleviating an important physical constraint that limits where salmon will spawn [57,58]. In addition, in-channel structures provide shear zones, which may be critical resting areas (energy refugia) for adult salmon [21].

Typically, in-channel structures are described as increasing habitat heterogeneity above riverine substrate (i.e., in-channel hydraulics); however, few studies describe the habitat variation that large woody debris, boulders, and other instream structure forms create within the hyporheic zone. In a laboratory setting, Thibodeaux and Boyle [22] precisely describe a complex flow pattern within a porous waveform using a dye-trace experiment. Analysis of stream and hyporheic water temperatures and computational fluid dynamics simulations have been used to demonstrate that in-channel rock vane structures induce hyporheic flow exchange [26]. In addition, a three-dimensional model developed by Tonina and Buffington [24] demonstrates that salmon redds themselves induce hyporheic exchange that is nested within the larger exchange patterns generated by pool–riffle topography. The localized subsurface flow patterns described by these studies were similar to the directional vertical hydraulic gradient pattern identified around large woody debris or boulders in our field study. The changes in horizontal and vertical surface water velocities adjacent to in-channel structures appeared to create larger pressure differences seen within the subsurface. The downwelling of surface water was evident at measurement locations placed just upstream of in-channel structure; however, the upwelling of hyporheic water

was apparent at measurement locations positioned downstream of in-channel structure. A similar hydraulic pattern was found within constructed steps in a lowland stream [59].

In contrast, the vertical hydraulic gradient fluctuated very little at measurement locations lacking in-channel structure providing further support that in-channel structure increases the natural variability within the subsurface environment, where salmonid embryos incubate. In addition, vertical hydraulic gradient magnitude was significantly increased at sites containing in-channel structure relative to sites lacking in-channel structure. Hester and Doyle [33] reported comparable findings with respect to instream steps, weirs and lateral structures and demonstrated that exchange could be maximized with reduced background groundwater discharge in low gradient streams. In some cases, this change may have a positive effect on incubating salmonid embryos. For example, the increased flushing of surface water through the hyporheic zone may diminish the effects of upwelling groundwater that can have a detrimental effect on incubating embryos [28,60]. Furthermore, our observation of generally lower dissolved oxygen levels within the substrate indicates increased downwelling may improve intergravel water quality. Similar positive benefits may be provided for developing embryos by carrying away metabolic wastes within the redd [61,62].

At most of our sites, intragravel conductivity measurements were similar to surface water values, indicating that surface water dominated the hyporheic zone. However, at Site 6, where in-channel structure was absent, hyporheic water conductivity was very high and dissolved oxygen levels were very low in comparison to all other sites, including where in-channel structure was present. It is likely that long residence groundwater, which contains more dissolved ions than surface water, was present in the hyporheic zone at this site given the high conductivity and low dissolved oxygen levels [46,53]. Interestingly, embryo survival, hatching rate, and growth were higher at Site 6 where in-channel structure was present. Although the shallow depths we studied were largely uninfluenced by long residence groundwater, in-channel structure appeared to improve the hyporheic conditions for salmonid embryos at Site 6 and may be beneficial to salmonid embryos in other places where long residence groundwater is common. Still, spatial and temporal variability of groundwater–surface water interactions, structure type and size, and hydrogeologic setting must be considered to determine the possible effects [58,60].

Overall, the embryo survival and hatching rates in our study were consistent with other studies using Chinook salmon embryos in the California Central Valley [40,63]. During a timeframe nearly identical to our study (December to January), Merz et al. [40] observed mean embryo survival rates of 22% in unenhanced and 29% in enhanced gravels of the LMR. Our study saw similar results at midstream Sites 4–6 ($\bar{x}$ = 25%), and at downstream Sites 7–9 ($\bar{x}$ = 23%), but lower survival at upstream Sites 1–3 ($\bar{x}$ = 6%). The higher hatching rates observed in our study ($\bar{x}$ = 44%, Sites 1–3) were similar to the hatching success of Chinook salmon embryos from the Sacramento River reared under full (100%) oxygen saturation, which was 35% at 10 °C and 45% at 14 °C [63]. The low embryo hatching rates observed in our study ($\bar{x}$ = 28%, Sites 7–9) were slightly higher than the hatching success of salmon embryos reared under hypoxic conditions (50% saturation) at 10 °C (21%) and at 14 °C (10%). The average daily hyporheic water temperatures recorded at Sites 1–9 in our study ranged from 8.6 to 13.8 °C and were similar to the temperature treatments examined by Del Rio et al. [63]. The dissolved oxygen levels recorded during our experiment appeared to be slightly lower than full saturation and slightly higher than 50% saturation at most study sites. However, we were not able to take measurements throughout the incubation period and were likely unable to capture the full range of conditions the embryos experienced.

At all upstream sites, in-channel structure presence/absence had no significant effects on Chinook salmon embryo survival and hatching rate. Upstream Sites 1–3 had poor embryo survival rates and there was a large discrepancy between survival and hatching rates, indicating that most observed mortality occurred after embryos hatched. This discrepancy between survival and hatching rates was also evident with the Hatchery

Control Group, which had the highest hatching rates (range = 64–75%), but a marked drop in overall survival ($\bar{x}$ = 27%). This was unexpected when compared with a Hatchery Control Group from a previous study [40], which observed a mean survival rate of 60%. Yet, water temperature, dissolved oxygen, and pH measurements at these sites were within acceptable ranges needed for proper Chinook salmon alevin development and survival [49–51], providing no clues to the causes of mortality.

However, results of other research suggests that there may be a tradeoff between rapid embryo development and growth and overall survival [64]. Consistent with our findings, other studies show that high temperature and dissolved oxygen results in a fast rate of development while low temperature and dissolved oxygen results in a slow rate of development [63,65]. Chinook salmon embryos recovered at upstream Sites 1–3 and the Hatchery Control Group were exposed to the warmest water temperatures and the highest dissolved oxygen levels and had the highest hatching rates and alevin total lengths among all Sites. However, the metabolic demand of embryos increases rapidly with temperature and development and late-stage embryos may experience oxygen limitations even when temperatures are considered suitable [64]. This relationship may help to explain the fast development yet low survival of embryos observed in the Hatchery Control Group and upstream Sites 1–3. Although hydrogen sulfide levels were not measured during this study, mats of rooted aquatic macrophytes were abundant in close proximity to upstream Sites 1–3. Growth and decay of aquatic vegetation surrounding salmonid spawning gravels may increase hydrogen sulfide in the hyporheic zone, which is toxic to salmonid embryos at low levels [66] and may have also contributed to the poor survival rates within Sites 1–3.

Unlike the upstream sites, there were significant increases in Chinook salmon embryo development within downstream Sites 7 and 9, where in-channel structure was present. Although not statistically significant, the same trend was evident at Sites 6 and 8. Vertical hydraulic gradient magnitude and dissolved oxygen were generally higher or similar at downstream sites containing in-channel structure present when compared with downstream site lacking structure. In addition, vertical hydraulic gradient magnitude was significantly increased at sites containing in-channel structure relative to sites lacking in-channel structure. Thus, the increased hatching rate of Chinook salmon embryos at downstream sites containing in-channel structure was likely due to changes in vertical hydraulic gradient direction, increased vertical hydraulic gradient magnitude, dissolved oxygen levels, or a combination of these variables. Consistent with results from other studies, poor water velocities and/or low dissolved oxygen levels appeared to delay the hatching times of Chinook salmon eggs [49,65]. To prevent death, embryos exposed to low oxygen levels will reduce their respiration rates and subsequently slow their growth and development rates [49]. These weaker embryos may not be able to withstand unfavorable riverine conditions over an extended time.

The variation in Chinook salmon alevin total length by site number may have been due, in part, to the insulating effect of Camanche Dam. As warmer water is released from the dam, it cools in a downstream progression due to the influence of ambient air temperatures during the cooler winter months [40]. In general, decreased alevin total length was evident at the downstream study sites. Furthermore, intergravel water temperature was included in the growth model and was positively correlated with alevin total length. According to Beacham and Murray [51] water temperature was a more important factor in determining Chinook salmon alevin length than egg size; however, alevin size decreased when incubation temperature was increased from 8 to 12 °C. In contrast, Merz et al. [40] found decreased growth of Chinook salmon embryos associated with cooler water temperatures downstream of Camanche Dam, as supported by our study.

The GLM for Chinook salmon embryo growth also showed a significant positive relationship between total length and pH. Newly hatched Chinook salmon alevins have been shown to be less tolerant of reduced pH levels than eyed eggs and developing fry [50,67]. In addition, after a 43-day period, Chinook salmon alevins reared at pH levels of 4.5, 5.0, and 5.5 were smaller in fork length than alevins reared at pH levels of 6.2

and 7.0, where temperature was held constant between groups [50]. Lower subsurface pH levels may identify areas with less hyporheic exchange due to the breakdown of organic material [46]. In addition, organic materials induce oxygen demands within the hyporheic environment, which may reduce the dissolved oxygen available to developing embryos [52]. Interestingly, when within-site comparisons were made, alevin total lengths at many downstream sites containing in-channel structure were significantly higher than lengths at paired sites lacking in-channel structure; however, vertical hydraulic gradient magnitude and dissolved oxygen were not included as significant variables in the growth model. Similar experiments examining water quality effects on salmon embryo survival and development have been conducted in a laboratory setting (62, 63, 65). These studies allow for full control of experimental conditions, which may lead to more clear results. However, field research is needed to contextualize the potential benefits of in-channel structure within the uncontrolled ecosystem. Despite some difficultly to control all experimental conditions, this study was important to challenge our hypotheses as they faced the realities of the natural environment.

It is important to note that this study did not examine in detail the extent to which in-channel structure influences spatial patterns of hydraulic flow paths and hyporheic flow patterns. Significant changes in vertical hydraulic gradient were detected at 22 cm below the gravel surface at distances between 0.1 m to 1.0 m from large woody debris or boulders, regardless of orientation. However, Hester and Doyle [33] reported that several factors can influence the magnitude of induced hyporheic exchange, including groundwater discharge rate, sediment hydraulic conductivity, structure size, structure type, depth to bedrock, and channel slope. In addition, Hester and Doyle [33] found that channel spanning structures were generally more effective in inducing hyporheic flow than were lateral structures. The three-dimensional extent to which in-channel structure affects the surface and hyporheic environments may be very important in determining the quantity of habitat altered. While the incubation depths we studied fell within the range of Chinook salmon redd burial depths, we did not examine the full range of possible incubation depths. Incubating Chinook salmon embryos may exist anywhere from 5 to 80 cm below the gravel surface depending on study river and specific location within the egg pocket [47]. Therefore, it is unclear if the relatively simple, isolated pieces of large woody debris and individual boulders used in our study would have induced changes in vertical hydraulic gradient magnitude beyond incubation depths of 22 cm. Moreover, the structure found on the LMR and used for this study tend to influence hydraulics only directly on the order of 5 to 15% of the channel width. It should also be noted that conditions within the manufactured embryo tubes may not reflect the natural hyporheic environment. It is possible water velocities in the tubes were reduced compared to the external natural substrate matrix. Further, because we used 14-day old embryos, not the entire incubation period, our results provide an index of survival. Even so, all embryos were exposed to the same tube effects, supporting the relative observations of this experiment.

Temporal variation was also not accounted for in our study and may play an important role in how in-channel structure affects the hyporheic environment. The warming effect of Camanche Dam on river water temperatures during the end of the spawning season is typically reversed in the early fall at the beginning of the spawning season. Merz and Setka [42] reported that hyporheic water temperatures are up to 4 °C higher than ambient temperatures in the early fall on the LMR. In-channel structure may be particularly important during this time frame, promoting the delivery of cooler water with higher oxygen saturation levels into the hyporheic zone; however, further investigation is warranted. Climate change may also lead to water temperature alterations in the LMR, further reducing suitable spawning habitats due to temperature exceedances, which may limit productivity in drought years. During this time these habitats may be improved by forcing elements, such as large woody debris or boulders, enhancing the exchange of cooler surface water to the subsurface. Although hydrological events are less common during the peak of Chinook salmon spawning season on the LMR, they have also been shown to have a considerable

effect on hyporheic conditions within a short time frame in other systems [68] and may diminish the effects of in-channel structure.

The increased habitat variation found around in-channel sites containing structure may be particularly beneficial to salmonids where spawning habitat is marginal. In undisturbed salmon streams, natural processes, including ample sediment supply, create habitat complexity that supports spawning and incubation habitats. For example, large boulders and other structures may not have been common in low gradient rivers in valley floors. Instead prior to dams, gravel supply would have been ample and channel width variations would create a majority of channel complexity. In the California Central Valley, very little salmon spawning habitat remains below dams, and what exists has been degraded by regulated stream flows, high water temperatures, lack of gravel recruitment, and sedimentation [69,70]. While the presence of in-channel structure may not improve the total surrounding area, some marginal habitats could be substantially enhanced. The upstream sites used for this experiment were located in enhanced high gradient spawning areas containing coarse substrate and few fines, while most of the downstream sites were located in low gradient spawning areas having a larger proportion of fines [40,41]. At downstream sites containing in-channel structure, Chinook salmon embryos had higher survival, hatching, and growth rates, relative to paired sites lacking in-channel structure. Salmonid embryo survival can be improved in high sand loading-mixtures through increased hydraulic gradient [56] and a strong association between Chinook salmon redds and large woody debris has been established in the downstream reaches of the LMR [18,19]. Results of our preliminary study support the idea that the presence of in-channel structure along with corresponding changes in physical spawning habitat may be particularly important in the marginal reaches of a lowland regulated stream. More research examining temporal variation and a full range of incubation depths is warranted, given the results of this preliminary research.

**Supplementary Materials:** The following supporting information can be downloaded at: https://www.mdpi.com/article/10.3390/w14010083/s1, Table S1: Correlation matrix among independent variables measured for Chinook salmon embryo survival, hatching rate, and growth at sites containing and lacking in-channel structure on the lower Mokelumne River. Variable abbreviations are as follows: COND = conductivity, DO = dissolved oxygen, SWD = surface water depth; SWV-A = surface water velocity (all); SWV-H = surface water velocity (horizontal); SWV-V = surface water velocity (vertical); VHG-D = vertical hydraulic gradient (directional measurements); VHG-M = vertical hydraulic gradient (magnitude); ATEMP = average daily temperature.

**Author Contributions:** Conceptualization, R.L.B. and J.E.M.; methodology, R.L.B. and J.E.M.; validation, R.L.B. and J.E.M.; formal analysis, R.L.B., J.E.M. and J.M.W.; investigation, R.L.B. and J.E.M.; resources, R.L.B. and J.E.M.; data curation, R.L.B.; writing—original draft preparation, R.L.B., J.E.M., and J.M.W.; writing—review and editing, R.L.B., J.E.M., and J.M.W.; visualization, R.L.B., J.E.M., and J.M.W.; supervision, J.E.M. All authors have read and agreed to the published version of the manuscript.

**Funding:** This research received no external funding.

**Institutional Review Board Statement:** The animal study protocol was approved by the Institutional Review Board of California State University, Sacramento (protocol code S06-007, 04/24/2006).

**Informed Consent Statement:** Not applicable.

**Data Availability Statement:** The data presented in this study are available in: Bilski, R.L. 2008. The effects of structural enhancement on Chinook salmon (Oncorhynchus tshawytscha) spawning habitat. M.Sc. Thesis. Department of Biological Sciences, California State University, Sacramento, CA, USA.

**Acknowledgments:** We thank M. Workman, J. Shillam, M. Saldate, C. Hunter, E. Rible, and many volunteers for field support. We thank the California State University, Sacramento Department of Biological Sciences Advisory and Graduate Committee members for helpful advice and comments during project development and implementation and three anonymous reviewers for helpful comments on a previous version of this manuscript. We gratefully acknowledge in-kind support from

East Bay Municipal Utility District. Additional support came from The Geology Department of California State University, Sacramento, and the University of California Davis Department of Air, Land, and Water Resources, which supplied field equipment. We thank four reviewers for constructive comments and suggestions on this manuscript.

**Conflicts of Interest:** The authors declare no conflict of interest.

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
