# Peer review of "Effects of In-Channel Structure on Chinook Salmon Spawning Habitat and Embryo Production"

_water, doi:10.3390/w14010083_

Round 1
Reviewer 1 Report
This study examines whether water flow regime alters salmon embryo survival in the presence or absence of structures in a stream. There are lots of data but not a clear story.
Experimental Design is not clear. If only 2 tubes are present per condition (structure or control) at each site, even if each tube contains 200 embryos it is still an N of 2, not much replication. What is between-tube variability? Also, normally the eggs would be open to the water above the gravel. Enclosing them in the tube could cause very serious changes including microorganism growth, current alteration, and stagnation. Also, what types of caps are on the ends of the tubes?
The authors should cite temperature vs development curves for salmon. Higher temp leads to faster development, but perhaps shorter embryos, how much shorter? Did their values match previous work?
Survival overall seems very low. Lines 339-347, none of these sites, including the controls, are very good at keeping the fish alive. Hatching rate and survival both seem low in all treatments. The numbers in Table 4 do not have much congruence with the numbers in the text for survival and hatching. This needs to be explained better. What kind of survival do the hatcheries here get as a rule?
Stream conditions are undergoing global temperature change. Many areas have restrictions or regulations on when dam water can be released and at what temperature. The lack of information on the time of day or changes due to dam water release is a problem.
All references are more than 10 years old – this is unacceptable. Please cite up-to-date sources.
Line 21, 64, italicize scientific names
Line 196 – where were measurements made, within the tube or in the surrounding waters?
Tables cross pages, and are hard to read – fix formatting of those in text, and move some to supplemental files (for example Tables 2, 3, 4, 6, 7).
Low DO, high conductivity at Site 6, not at others – why? Table 5
The topic is interesting but the presentation is not convincing.
Reviewer 2 Report
GENERAL COMMENTS
The manuscript focuses on an interesting topic of the effects of in-channel structure on the spawning habitat and embryo production of Chinook salmon. This is indeed important for the management of salmonid populations, and I think this could be better explored in the text, particularly at the end of the Discussion and Abstract (a “take-home” message would be important to increase the appealing of the manuscript). The text is overall well-written and structure, with an adequate methodology. My major concerning relies on the description of your control and treated experiences (see comment of lines 151-152). What was the sample sizes of your sites? As far as I understood, did you employ one 1 (one) boulder (or 2) per site? I am not sure if I understand correctly, but this seems a too small scale and not representative, and believe some clarifications are needed. Other comments are listed below, should the authors want to use them to improve their manuscript.
SPECIFIC COMMENTS
L21 – Scientific name in italics
L44 – replace ; by ,
L85-87 – Could you provide some references?
L108 – 263 m3/s 142 ?
L133- What was the river gradient in the 3 sections?
Table 1 – What do these numbers represent, i.e. 1 boulder, 2 boulders, etc? you only placed 1, 2 boulders in your study sites? Please clarify.
Table 1 – I do not understand how you have 18 sites here (9 x 2 (control + restored)) on only 9 on figure 1 (map)? What’s the distance (in m) between a pair of control/restored? I presume that are very, very close to each other.
L150 – “another nearby site”- How near? Be specific.
L151-152 – “To be defined as a site with in-channel structure, they needed to contain at least one in-channel structure feature”. You mean only 1 boulder? 2 boulders? In a river segment? But how long are your sites? This is quite tiny.
L156 – Did your sites had flow deflectors? Provide details.
L163-164 - “A total of 8 egg tubes was placed around each site, two at each measurement location”. What for??? Did was not described before, so please provide details on the text.
L168 – Which ones? Provide references.
L172- How many measurements did you perform per site?
L183- How many sampling periods did you have? When?
L286 – What was the cut-off point? R> |0.70|? Other?
L293 and throughout the manuscript – site with small s
Table 2 – please check “measure-“. It seems the rest of the word is missing.
Table 4, left column – Each site corresponds to a pair of sites (i.e. control and treatment), correct? This should be clearer on the table.
L352-356 – Did you get significant differences for the different sites? If so, this should be marked at each of the box and whisker graph.
L359 – SE? (standard error?). Full name upon first citation.
L393 – I believe you mean |0.6| (i.e. whether positive or negative) instead of 0.6?
L404 – matrix instead of martrix.
L410 – You should also refer why some of the correlations are marked in bold. The same for Table 7.
L423 – “acceleration”? Perhaps “increases would be a better word.
L593 – What about practical implications for management? I would like the authors to comment on this.
Reviewer 3 Report
The research design was interesting.
However, it seems that a number of parameters that have not been taken into account may ultimately be responsible for the aquired results such as the sediment structure or the fluctuations of the values of the water's physicochemical parameters during the experiment.
No information is given on the composition of the substrate which should have also been taken into account.
There is also no information on the frequency of the temperature or velocity measurements. This has to be added.
Laboratory experiments that would allow full control of all conditions would show in a clearer way the effect of parameters such as velocity, pH and Temperature on survival rates.
The above should be discussed in more detail.
More comments are provided in the attached pdf file.
Also please check for the journal’s policy concerning the use of live fish and their killing for experimental issues.

Reviewer 4 Report
The draft manuscript is well written and the results will be of strong interest to fisheries managers and stream restoration practitioners. I have a small number of comments, but ones that likely require the authors to use a different/additional statistical tests.
Materials and Methods:
lines 173-174. How does the timing of flow measurements relate to the period of staging or spawning by Chinook salmon?
lines 258-262. Authors need to demonstrate that 3-way ANOVA is an appropriate test. The sampling design is spatial at the site and study system scale - therefore, one can expect correlated dependent variables and an underlying structure along the river reach. This increases the risk of Type 1 error. At a minimum, an assessment of spatial autocorrelation should be undertaken and results provided. Still, I am not sure how one could consider any of the site-level measurement locations (upstream, downstream and lateral) to be independent of the other 3 locations.
I recommend that the authors evaluate whether: 1) MANOVA (with subsequent 1-way ANOVAs) provides an alternative approach to address correlated dependent variables; or 2) identify ANOVA designs accommodate the correlation of spatial units within sampling sites.
Discussion
There is a lot of material in the discussion, I recommend that the authors use sub-headings to break the section up into separate topics.
Round 2
Reviewer 1 Report
The content is much clearer with the revision. The asterisks on the figures help. On Fig. 5C, it looks to me like Site 3 has within-site differences, but not Site 4. Is this marked in error?
The added citations strengthen the discussion.
Author Response
Response to Reviewer 1 Comments
Point 1: The content is much clearer with the revision. The asterisks on the figures help. On Fig. 5C, it looks to me like Site 3 has within-site differences, but not Site 4. Is this marked in error?
Response 1: This is not marked as an error, but the mean is not provided in this Figure (and others) and may be helpful to the reader. Site 3 is not statistically significant at the 0.05 level, but Site 4 is significantly different, as indicated in Table 4. The authors checked the data to confirm the results were accurate and to confirm use of the correct data for the Figures. We updated Figures 3 through 5 to include mean values in the box plots, which will help with interpretation of the results.
Reviewer 2 Report
Dear authors,
I am overall happy with this revised version as well as with the answers provided to my previous concerning. No doubt the manuscript has improved significantly over the original version.
Author Response
We thank you again for the helpful comments and suggestions, which have significantly improved our manuscript.